# Genetic Diversity of Virulent Polymyxin-Resistant *Klebsiella aerogenes* Isolated from Intensive Care Units

**DOI:** 10.3390/antibiotics11081127

**Published:** 2022-08-19

**Authors:** Kesia Esther da Silva, Gleyce Hellen de Almeida de Souza, Quézia Moura, Luana Rossato, Letícia Cristina Limiere, Nathalie Gaebler Vasconcelos, Simone Simionatto

**Affiliations:** 1Laboratório de Pesquisa em Ciências da Saúde, Universidade Federal da Grande Dourados, Dourados 79804-970, MS, Brazil; 2Division of Infectious Diseases and Geographic Medicine, Stanford University, Stanford, CA 94304, USA; 3Instituto Federal do Espírito Santo, Campus Vila Velha, Vila Velha 29106-010, ES, Brazil; 4Laboratório de Microbiologia, Hospital Universitário da Universidade Federal da Grande Dourados—HU/UFGD/EBSERH, Dourados 79823-501, MS, Brazil

**Keywords:** colistin-resistance, genetic mechanisms, whole genome sequencing, *Enterobacteriaceae*, intensive care unit, ICE*Kp*

## Abstract

This study evaluated the scope and genetic basis of polymyxin-resistant *Klebsiella aerogenes* in Brazil. Eight polymyxin-resistant and carbapenemase-producing *K. aerogenes* strains were isolated from patients admitted to the ICU of a tertiary hospital. Bacterial species were identified by automated systems and antimicrobial susceptibility profile was confirmed using broth microdilution. The strains displayed a multidrug resistant profile and were subjected to whole-genome sequencing. Bioinformatic analysis revealed a variety of antimicrobial resistance genes, including the *bla*_KPC-2_. No plasmid-mediated colistin resistance gene was identified. Nonetheless, nonsynonymous mutations in *mgrB, pmrA, pmrB,* and *eptA* were detected, justifying the colistin resistance phenotype. Virulence genes encoding yersiniabactin, colibactin, and aerobactin were also found, associated with ICE*Kp4* and ICE*Kp10*, and might be related to the high mortality observed among the patients. In fact, this is the first time ICE*Kp* is identified in *K. aerogenes* in Brazil. Phylogenetic analysis grouped the strains into two clonal groups, belonging to ST93 and ST16. In summary, the co-existence of antimicrobial resistance and virulence factors is deeply worrying, as it could lead to the emergence of untreatable invasive infections. All these factors reinforce the need for surveillance programs to monitor the evolution and dissemination of multidrug resistant and virulent strains among critically ill patients.

## 1. Introduction

*Klebsiella aerogenes* represents one of the bacterial species causing nosocomial infections; it also causes respiratory infections, urinary tract infections, endocarditis, skin or soft-tissue infections, abdominal infections, and osteomyelitis [1,2,3]. Moreover, it easily acquires numerous genetic mobile elements containing resistance and virulence genes that increase its pathogenicity and lethality. Therefore, this group of pathogens is receiving special attention in clinical practice [4,5].

Polymyxin-resistant *Enterobacteriaceae* is an urgent threat to human health because polymyxin is regarded as a drug of last resort in the therapeutic management of Gram-negative infections that otherwise have limited or no therapeutic options [6]. Resistance to colistin is typically caused by lipopolysaccharide (LPS) modification with 4-amino-4-dexoxy-L-arabinose or phosphoethanolamine. These positively charged molecules reduce the overall negative charge of LPS, leading to a smaller electrostatic interaction with colistin and preventing cell lysis [7]. In addition to colistin resistance due to chromosomal mutations, plasmid-borne colistin resistance has been reported; all these contribute to decreased binding of colistin to LPS [8]. Furthermore, *Klebsiella* spp. may harbor several virulence factors associated with survival and pathogenesis, including a e integrative conjugative element (ICEs), which is a group of mobile genetic element transferable between bacterial species, associated with hypervirulence in *K. pneumoniae* strains (e.g., ICE Kp10 encoding the siderophore yersiniabactin and genotoxin colibactin) [9,10,11,12].

Effective treatment of infections caused by these pathogens as well as the implementation of adequate preventive measures to effectively contain the spread remain challenging [13]. Various studies have investigated the mechanisms associated with polymyxin-resistance in Gram-negative infections. The polymyxin exposure may triggered the genetic events that lead to gene modifications in the polymyxin-resistant isolates [14,15,16,17,18,19,20,21,22]. However, studies on polymyxin-resistant *Enterobacter* spp. have been limited compared with those on other Enterobacteriaceae pathogens. In this study, we identified the genetic mechanisms associated with polymyxin resistance in *K. aerogenes* isolates from a tertiary hospital located in the Central-Western region of Brazil by using whole-genome sequencing.

## 2. Results and Discussion

### 2.1. General Patient Characteristics

Over the study period, a total of 340 cultures were performed and 124 cases of *Enterobacteriaceae* infection were identified. Twenty-five strains were isolated from patients presented to the adult intensive care unit (ICU). Of these, polymyxin-resistant and carbapenemase-producing *K. aerogenes* strains were isolated from eight patients. The majority of patients were female (62.5%), with a median age of 45 years (range, 31–76 years). No significant differences (*p* > 0.05) in baseline demographics were observed among the patients. All patients had a diagnosis of infectious diseases at the sample collection time. Four of them had bloodstream infections, whereas the remaining patients had a diagnosis of pulmonary and urinary infections. All patients were exposed to antibiotics 30 days prior to the initial detection of the polymyxin-resistant strain (Appendix A Table A1). Outcome analysis revealed that five patients infected with polymyxin-resistant *K. aerogenes* died of sepsis, giving an infection-attributable high mortality rate of 62.5% (Appendix A Table A1).

However, we could not identify any other article that reported the mortality rate for patients with polymyxin-resistant *K. aerogenes*. Some studies have evaluated patients with carbapenem-resistant *K. aerogenes* infection and reported mortality rates of 20% [23], 37.5% [24], and 40.3% [25] for patients with severe sepsis. These results reinforced the criticality of infection control measures to prevent the spread of such bacteria in hospitals. 

Regarding phylogenetic analysis and virulome, the core genome phylogeny and MLST analysis identified similar populations and were used to classify the strains into two strongly clonal groups. On the basis of SNPs, a maximum-likelihood phylogenetic tree was constructed, demonstrating that strains were closely related and partitioned into two clades. MLST analysis identified two different sequence types: ST93 (*n* = 5), belonging to clonal complex 3 (CC3), and ST16 (*n* = 3). ST93 has been previously described in clinical isolates of *K. aerogenes* in Brazil [26] and, at the moment, it represents one of the dominant global *K. aerogenes* clones associated with human infections [11]. On the other hand, to date, there is a single register of ST16 in a clinical isolate from China, from 2014, according to data available at the PubMLST database (https://pubmlst.org/bigsdb?db=pubmlst_kaerogenes_isolates&page=query, accessed on 18 May 2022). The high genetic similarity of strains suggested transfer events between patients. Therefore, measures must be implemented to avoid nosocomial transmission [27].

Virulome analysis revealed the presence of genes encoding yersiniabactin, colibactin, and aerobactin. Additionally, distinctive yersiniabactin (*ybt*) lineages (*ybt* 0, *ybt* 10, and *ybt* 17) were found associated with two different structural variants of ICE*Kp* (ICE*Kp4* and ICE*Kp10*) (Figure 1). ICE*Kp* is the most common mobile genetic element associated with virulence in *K. pneumoniae* isolates, facilitating the spread of virulence genes within the population [9]. To the best of our knowledge, until now, there is only one report of ICE*Kp* in *K. aerogenes* isolates from New York, USA, which was even identified as ICE*Kp10* [11]. In fact, ICE*Kp4* and ICE*Kp10* seems to be the most widely distributed variants [9]. As ICE*Kp* elements harbor yersiniabactin and colibactin genes, which are considered key bacterial virulence factors, we could even hypothesize that a higher severity of infection with death outcome, as observed in some patients, could be related to the presence of these elements.

### 2.2. Antimicrobial Susceptibility Profile, Resistome and Plasmid Incompatibility Groups

Regarding the susceptibility profile (Appendix A Table A2), all strains exhibited resistance to polymyxin B (MIC_50_ ranging from 8 mg/L to 32 mg/L). Resistance gene profiles varied among strains, with some harboring several beta-lactam, aminoglycoside, tetracycline, quinolones, sulfonamides, trimethoprim, amphenicol, fosfomycin, macrolide and rifamycin resistance determinants. The *bla*_KPC-2_ and *bla*_OXA−1_ carbapenemase genes were identified in all isolates. In addition, the strains carried the *bla*_CTX−M−15_ (75%, *n* = 6), *bla*_TEM-1B_ (75%, *n* = 6) and *bla*_OXA−9_ groups (12.5%, *n* = 1) extended-spectrum β-lactamase (ESBL) encoding genes. Although it was not possible to determine the location of these antimicrobial resistance genes, six different plasmid replicons were identified among these isolates (Figure 1). This finding suggests the horizontal transmission of at least some of these genes, mainly those encoding resistance to beta-lactams, once they are frequently harbored by plasmids [28].

Plasmid-encoded colistin resistance genes (*mcr*-like) were not detected. Nonetheless, deleterious point mutations were found in *mgrB*, *pmrA*, *pmrB*, *eptA*, and *arnT* genes (Table A2). All strains exhibited alterations in the *mgrB.* The alterations were located in two different amino acids positions (M1V and G37S). A second potential mechanism of mutational colistin resistance, due to a substitution at amino acid position (T296S) in *eptA*, was identified in all ST93 isolates (62.5%). A third potential mechanism of mutational resistance to colistin, due to an amino acid position substitution in *pmrAB*, was observed in the ST16 isolates (37.5%).

Studies have suggested that alterations in *mgrB* and *pmrAB* may be responsible for polymyxin resistance in Gram-negative pathogens [13,29,30,31]. The inactivation of *mgrB*, which encodes a negative feedback regulator of the PhoQ-PhoP signaling system, was recently demonstrated to be a common mutational mechanism responsible for acquired polymyxin resistance among the clinical isolates of *K. pneumoniae*, *Enterobacter* spp., and *E. coli* [14,15,21,32,33,34,35]. Mutations in *mgrB* may be the main determinant for colistin resistance in *K. aerogenes* [19]. Researchers from Croatia have reported that polymyxin resistance in *K. aerogenes* occurred due to *mgrB* present in a wild-type sequence. However, they did not detect the presence of *mcr*-1 or *mcr*-2 plasmid genes [36].

Polymyxin resistance is most commonly regulated by two-component systems, including PmrAB and PhoPQ [17]. Alterations in *pmrAB* or *eptA* may be a mechanism of colistin resistance, as described among *E. coli* strains [37]. A French study reported that a G53S substitution in PmrA resulted in a polymyxin resistance phenotype in a *K. aerogenes* strain [38]. Mutations in *pmrB* were associated with colistin resistance in *A. baumannii* [16]. To the best of our knowledge, it is the first time that the mutation of *mgrB*, responsible for colistin resistance in *K. aerogenes,* has been detected in Brazil.

Our findings demonstrated that the mechanisms of polymyxin resistance in *K. aerogenes* appear to be highly diverse. The emergence and spread of polymyxin-resistant strains have been reported, especially in the hospitals where *bla*_KPC-2_ is endemic, and the increased consumption of polymyxins has been proven to be a major risk factor for polymyxin-resistance development [15,18,20]. Previous studies have reported that high levels of antimicrobials, including polymyxin, are frequently administered in Brazilian ICUs, mainly after bacterial isolates have become resistant to almost all other available antibiotics [39]. We hypothesized that the polymyxin exposure triggered the genetic event that led to gene modifications in the first isolate of each clone. However, because of the very recent release of *K. aerogenes* MLST profile, there is no sufficient information about the clonal characteristics and global distribution of this lineage. This study is limited by its small sample size; it only included the polymyxin-resistant *K. aerogenes* strains isolated during the study period. However, the emergence and clonal spread of mutational colistin resistance mediated by three distinct mechanisms over the course of three months is concerning, especially for patients admitted in ICUs.

## 3. Materials and Methods

### 3.1. Bacterial Strains

Polymyxin-resistant *K. aerogenes* recovered from patients hospitalized at a public tertiary care hospital, in Brazil, from August 2016 to October 2016 were included (Figure 2). Samples were collected by hospital nurses as part of routine screening used in patient diagnosis. Patient characteristics along with clinical and demographic data were reviewed and entered into Research Electronic Data Capture (Redcap) (Vanderbilt University, Nashville, TN, USA). Data regarding the clinical outcome were reviewed. Death due to any cause or death attributable to infection was assessed. Septic shock was defined as sepsis associated with organ dysfunction, accompanied by persistent hypotension following volume replacement.

### 3.2. Bacterial Identification and Antimicrobial Susceptibility Testing

Identification and screening of antimicrobial-resistant bacterial species were performed using Phoenix^®^ Automated System (BD Diagnostic Systems, Sparks, MD, USA) according to the manufacturer’s instructions. After isolation, the susceptibility profile was confirmed and minimal inhibitory concentrations (MICs) of antimicrobials were determined using broth microdilution, following the Clinical and Laboratory Standards Institute guidelines [40]. The following classes of antimicrobials were tested: Cephalosporins (cefotaxime, ceftazidime, ceftriaxone and cefepime), carbapenems (ertapenem, imipenem and meropenem), fluoroquinolones (ciprofloxacin and levofloxacin), aminoglycosides (amikacin and gentamicin), monobactams (aztreonam), and polymyxins (polymyxin B). Multidrug resistant were defined as resistance to one or more antimicrobials from three or more tested categories [41].

### 3.3. Whole-Genome Sequencing (WGS)

Genomic DNA was extracted from fresh cultures by using QIAamp^®^ DNA Mini Kit (Qiagen, Hilden, Germany). The concentration and purity of DNA were determined using a Qubit^®^ 2.0 fluorometer and the dsDNA BR Assay Kit (Life Technologies, Carlsbad, CA, USA). Sequencing libraries were prepared using the Nextera library kit (Illumina, San Diego, CA, USA). The prepared libraries were sequenced with 150 bp paired-end reads via IlluminaMiSeq Platform (Illumina, San Diego, CA, USA), as described in a previous study [42]. FastQC 0.11.2 was used to preprocess the reads [43]. Each read set was assembled using SPAdes 3.6.1 [44] with k-mer sizes of 21, 33, 55, 77, 99, and 127 and mismatch correction. These sequences were annotated using Prokka [45]. Species identification was performed using Kraken [46]. The *K. aerogenes* core genome was defined as the concatenation of coding sequences presenting one copy in all final assemblies [47]. The whole-genome sequences described in this paper have been deposited in ENA (European Nucleotide Archive) (Project: PRJEB25746; accession numbers in Appendix A Table A3).

### 3.4. Bioinformatics Analysis

Single nucleotide polymorphisms (SNPs) were identified through mapping of Illumina reads to a reference genome (*Klebsiella aerogenes* ATCC 13048). Maximum likelihood phylogenetic trees were constructed using RAxML 8.1.23 [48]. Analyses were performed with 100 bootstrap replicates per run, with a generalized time-reversible model and a gamma distribution to model site-specific rate variation (GTR+Γ substitution model; GTRGAMMA in RAxML); final visualization was performed using FigTree 1.3.1. We selected a single tree with the highest maximum likelihood as the best tree. For the larger tree containing global isolates, clades collapsed manually in R. SRST2 [49] were used to map known alleles and identify multilocus sequence typings (MLSTs) directly from reads according to the *K. aerogenes* MLST database [50].

Assembled genomes were submitted to ResFinder 4.1 (https://cge.cbs.dtu.dk/services/ResFinder/, accessed on 6 February 2022) for prediction of acquired antimicrobial resistance genes. Chromosomal genes *mgrB*, *phoP*, *phoQ*, *pmrA*, *pmrB*, *eptA*, and *arnT* were manually screened for point mutations associated with colistin resistance by using blastn and blastx tools, and *Klebsiella aerogenes* ATCC 13048 was used as reference genome (GenBank accession number QVMZ00000000.1). PROVEAN v. 1.1.3 software was used to predict the functional effect of each found mutation, considering deleterious mutations as related to the resistant phenotype [51]. Plasmid replicon sequences were identified using ARIBA to screen reads for replicons in the PlasmidFinder database [52]. Virulence genes were identified by comparison of the assembled genome with genes from the Virulence Factor Database (http://www.mgc.ac.cn/VFs/main.htm, accessed on 6 February 2022) and ICE*Kp* variants were determined by using blastx. A ≥98% threshold for sequence identity was used for resistance and virulence genes identification (Figure 1).

## 4. Conclusions 

Our findings demonstrated that chromosomal alteration in *mgrB* was the primary reason behind polymyxin-resistance development. To the best of our knowledge, so far, this is the first report identifying ICE*Kp* in isolates of *K. aerogenes* in Brazil, namely ICE*Kp4* and ICE*Kp10*. A careful and continued surveillance system providing epidemiological and molecular information is required to follow the evolution of polymyxin resistance in Brazil and to limit the risk of outbreaks caused by these high-risk clones.

## Figures and Tables

**Figure 1 antibiotics-11-01127-f001:**
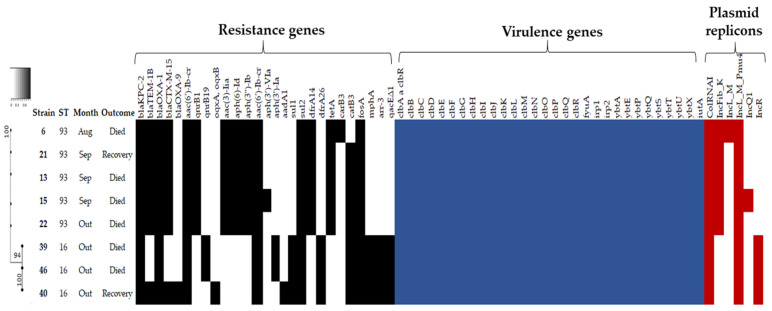
Maximum-likelihood phylogenetic tree and gene content of polymyxin-resistant *K. aerogenes* strains isolated from ICU. Black bars represent the presence of resistance genes, blue bars the virulence genes and red bars the plasmid profile, predicted by the ResFinder 4.1 program.

**Figure 2 antibiotics-11-01127-f002:**
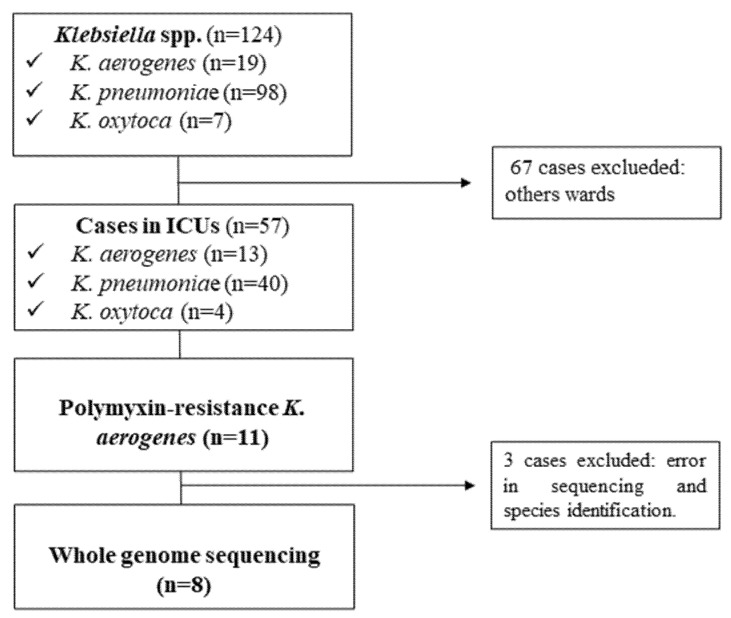
Flowchart of the study design.

## Data Availability

The whole-genome sequences described in this paper have been deposited in ENA (European Nucleotide Archive) (Project: PRJEB25746) at https://www.ebi.ac.uk/ena/browser/view/PRJEB25746?show=reads (accessed on 27 May 2018).

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
