# Peer review of "Genetic Diversity of Virulent Polymyxin-Resistant Klebsiella aerogenes Isolated from Intensive Care Units"

_antibiotics, 2022, doi:10.3390/antibiotics11081127_

Round 1

Reviewer 1 Report

The manuscript is well designed and shares interesting findings. Any surveillance study to control and check MDR issue is always helpful. I guess if the authors can add some more isolate from recent time it will strengthen the paper.

Author Response

Reviewer: 1

Comments to the editor and authors

The manuscript is well designed and shares interesting findings. Any surveillance study to control and check MDR issue is always helpful. I guess if the authors can add some more isolate from recent time it will strengthen the paper.

Response: We appreciate and agree with the reviewer's comment. The small number of our collection might be explained by the inclusion criteria of our study. The inclusion criteria were the isolation of a polymyxin-resistant K. aerogenes organism from patients admitted to Intensive care units. The small number of strains included may have impacted the significance of our findings. This was a limitation in our study. Nonetheless, more epidemiological surveillance studies need to be carried out in order to promote surveillance programs to monitor the evolution and spread of polymyxin-resistant K. aerogenes strains among critically ill patients.

Your Sincerely,

Simone Simionatto, PhD

Reviewer 2 Report

I would like to thank the authors for providing important information for the investigation of the genetic diversity of virulent polymyxin B-resistant Klebsiella aerogenes from ICUs in Brazil. They used whole genome sequencing to determine the virulence determinants and antimicrobial resistance profiles. The most interesting finding is that this is the first study in Brazil to identify ICEKp in K. aerogenes. I believe the findings of the present study will assist in the prevention and control of MDR pathogens in patients.

However, I have a few comments regarding this manuscript, as follows:

Comments

Introduction

Line 48-52: This sentence seems complicated, please rephrase it.

Results and Discussion

Ø  Is it possible to split this section into two separate sections: “Results” and “Discussion”?

Ø  HHow many virulence and resistance genes (total) were detected in each strain using WGS? Is it possible to mention their numbers (using the Virulence Factor Database and ResFinder Database, respectively)?

Ø  I’m just wondering if it is possible to show identity percent among isolated eight K. aerogenes using a table or figure, even as supplementary material.

Line 66-73: All the eight isolates were polymyxin B resistant, but according to Table A1, only two patients were exposed to polymyxin B. But why did all the strains show resistance to polymyxin B? due to environmental transmission? horizontal gene transmission? or others? Please discuss the reasons here.

Line 79: Please move the reference [25] after 40.3%, e.g., 40.3% [25] for patients………

Line 99: Please replace “providing” with “facilitating”

Line 109-110: According to the CLSI 2020, the resistant MIC value of amikacin is ≥64 µg/mL (mg/L). But according to Table 2A, three isolates showed a MIC value of 32 mg/L. So, they should be intermediate in nature. Then, how could you state that all the strains showed high resistance rates against the tested antimicrobials? Did you assume intermediate isolates as resistant? If yes, why? Also, please mention it in the materials and methods section. If not, please correct it.

Materials and Methods

Line 165: Were they the only polymyxin-resistant isolates? In the abstract, you mentioned, that isolates were both polymyxin-resistant and carbapenemase-producers. Please clarify it.

Line 165-166: Did your team recover those isolates by yourselves, or did you just collect those isolates from the hospital? Please clarify it. Also, by which methods did they or you recover those polymyxin-resistant and carbapenemase-producing K. aerogenes isolates? If there is any published data on those isolates, please cite that reference. If not, please briefly describe the methods of their isolation here.

Line 168: Please mention the developer or company name of Redcap software, as well as the city and country name. Also, what kind of analysis did you perform with the patients’ characteristics, clinical, and demographic data? Please mention it here.

Line 174: Please mention the names and classes of antimicrobials here.

Line 176: As you mentioned, MDR isolates were subjected to the WGS, please define “multi-drug resistant” isolates here [after reference 40] with reference.

Tables and Figures

Line 248: Table 2A- Please mention which were resistant, intermediate, or sensitive, with their MIC value.

Line 274: Figure 2A: Here “R” means “polymyxin-resistant”? If yes, how did you select “11” as polymyxin-resistant K. aerogenes? According to your figure, the number of “R” is 12. And if “R” is not polymyxin-resistant, please clarify what it is.

Author Response

15th August 2022.

Dear Prof. Dr. Nicholas Dixon,

Thank you very much for inviting us to submit a revised version of our manuscript entitled: “Genetic diversity of polymyxin-resistant Klebsiella aerogenes isolated from Intensive care unit in Brazil. Indeed, the reviewers have raised a number of important concerns. We have now made a thorough revision of the manuscript taking into account these points, which helped in improving the quality of the manuscript. Please find bellow a point-by-point response to the reviewers and their comments.

Reviewer 2

Comments to the editor and authors.

Line 48-52: This sentence seems complicated, please rephrase it.

Response: We appreciate the reviewers' constructive comments to substantially improve the readability of our manuscript. We apologize for the confusion and have changed the text for a better understanding (lines 48-53).

Is it possible to split this section into two separate sections: “Results” and “Discussion”?

Response: We appreciate the reviewers' comments. The manuscript format meeting the standards allowed by the journal. Due to the reduced time for the reviewed, we would be grateful if we could keep the manuscript in this format.

How many virulence and resistance genes (total) were detected in each strain using WGS? Is it possible to mention their numbers (using the Virulence Factor Database and ResFinder Database, respectively)?

Response: We appreciate the reviewer’s comment. A ≥98% threshold for sequence identity was used for resistance and virulence genes identification. We have listed only the genes that were identified in our analysis. This information was added in the text for a better understanding (lines 232-233) and in the Figure 1.

I’m just wondering if it is possible to show identity percent among isolated eight K. aerogenes using a table or figure, even as supplementary material.

Response: We have made these suggested changes to Figure 1.

Line 66-73: All the eight isolates were polymyxin B resistant, but according to Table A1, only two patients were exposed to polymyxin B. But why did all the strains show resistance to polymyxin B? due to environmental transmission? horizontal gene transmission? or others? Please discuss the reasons here.

Response: We appreciate constructive comments from reviewers. Indeed, only two strains were isolated from patients who had already been exposed to polymyxin. Polymyxin resistance in these strains may have been induced by selection pressure. However, the acquisition of resistance is a multifactorial and complex process. In this study, the mechanisms of polymyxin resistance in these K. aerogenes isolates included deleterious point mutations in the mgrB, pmrA, pmrB, eptA and arnT genes (Table 3.A). Plasmid-encoded resistance (mcr-like) were not detected. All strains exhibited changes in mgrB. The changes were located at two different amino acid positions (M1V and G37S). A second potential mechanism of mutational resistance to colistin, due to substitution at the amino acid position (T296S) in eptA, was identified in all ST93 isolates (62.5%). A third potential mechanism of mutational resistance to colistin, due to an amino acid position substitution in pmrAB, was observed in the ST16 isolates (37.5%) (Lines 126-133).

Line 79: Please move the reference [25] after 40.3%, e.g., 40.3% [25] for patients………

Response: Following the reviewer’s suggestion, we have revised the reference (line 82).

Line 99: Please replace “providing” with “facilitating”

Response: We are grateful about the reviewers comment and the word was replaced (line 102).

Line 109-110: According to the CLSI 2020, the resistant MIC value of amikacin is ≥64 µg/mL (mg/L). But according to Table 2A, three isolates showed a MIC value of 32 mg/L. So, they should be intermediate in nature. Then, how could you state that all the strains showed high resistance rates against the tested antimicrobials? Did you assume intermediate isolates as resistant? If yes, why? Also, please mention it in the materials and methods section. If not, please correct it.

Response: We thank the reviewer for the comment and apologize for the confusion. The information has been revised and corrected in the text (lines 113-115).

Line 165: Were they the only polymyxin-resistant isolates? In the abstract, you mentioned, that isolates were both polymyxin-resistant and carbapenemase-producers. Please clarify it.

Response: We appreciate constructive comments from reviewers and apologize for the confusion. The inclusion criteria of this study were the isolation of a polymyxin-resistant K. aerogenes strain from patients admitted to Intensive Care Units. Subsequently, we performed genome sequencing, identified as a result the mechanism of polymyxin resistance and that all strains were carbapenemases producers (e.g. blaKPC-2 and blaOXA−1). We have included this information in the text (lines 118-119).

Line 165-166: Did your team recover those isolates by yourselves, or did you just collect those isolates from the hospital? Please clarify it. Also, by which methods did they or you recover those polymyxin-resistant and carbapenemase-producing K. aerogenes isolates? If there is any published data on those isolates, please cite that reference. If not, please briefly describe the methods of their isolation here.

Response: This is a point well-taken and we appreciate the reviewers’ comment. Samples were collected by hospital nurses as part of routine screening used in patient diagnosis. Then isolation and identification of the bacterial species by using the Phoenix® Automated System (BD Diagnostic Systems, Sparks, MD) by the hospital's microbiology laboratory staff. Subsequently, the strains were forwarded to our research group and included in the study. Following the reviewer’s suggestion to improve the quality of our manuscript we changed the text for a better understanding (lines 172-173).

Line 168: Please mention the developer or company name of Redcap software, as well as the city and country name. Also, what kind of analysis did you perform with the patients’ characteristics, clinical, and demographic data? Please mention it here.

Response: We appreciate and agree with the reviewer's correction, we added the name of the Redcap software developer (lines 174-175). Descriptive analysis of patient characteristics and clinical data on hospitalization are shown in Table 1A.

Line 174: Please mention the names and classes of antimicrobials here.

Response: In response to the reviewer’s comment, this information was added in the Materials and Methods section (lines 179-182).

Line 176: As you mentioned, MDR isolates were subjected to the WGS, please define “multi-drug resistant” isolates here [after reference 40] with reference.

Response: We apologize for the confusion and have added the definition to the text for better understanding (lines 190-191).

Tables and Figures

Line 248: Table 2A- Please mention which were resistant, intermediate, or sensitive, with their MIC value.

Response:  The suggestion was accepted.

Line 274: Figure 2A: Here “R” means “polymyxin-resistant”? If yes, how did you select “11” as polymyxin-resistant K. aerogenes? According to your figure, the number of “R” is 12. And if “R” is not polymyxin-resistant, please clarify what it is.

Response: We apologize for the confusion and have revised Figure 2. We removed information on susceptibility and resistance to carbapenems.

Your Sincerely,

Simone Simionatto, PhD

Reviewer 3 Report

     The authors isolated and characterized Klebsiella aerogenes from hospitalized patients in Brazil. Considering the severe health impacts of the virulent and antibiotic resistant K. aerogenes, the current study is highly relevant to the public. The authors did a great job in explaining the details on the significance of the research output. Minor corrections should be addressed prior to publication. Those are listed below

1.     Correct spelling - Line 23 (high), Figure 2A (excluded)

2.     Please clarify - what is meant as literature searching in Fig 2A caption; did the authors mean hospital record search or from articles that were previously published?

3.     Please eliminate repeating sentences in lines 182-183.

4.     Please give the full form for SNP and ESBL wherever it was first given in the text

5.     Where any other bacterial agents isolated from the patients from the clinical samples (along with K. aerogenes)? How was it confirmed that the cause of death was solely due to K. aerogenes?

6.     Please consider adjusting space between the subtitles and a new paragraph (eg. Line 108 and 109)

7.     Please explain Figure 1A – what does the number in the first column represent (before ST)? 

8.     Consider moving the hypothesis from the results and discussion to the introduction section

Author Response

15th August 2022.

Dear Prof. Dr. Nicholas Dixon,

Thank you very much for inviting us to submit a revised version of our manuscript entitled: “Genetic diversity of polymyxin-resistant Klebsiella aerogenes isolated from Intensive care unit in Brazil. Indeed, the reviewers have raised a number of important concerns. We have now made a thorough revision of the manuscript taking into account these points, which helped in improving the quality of the manuscript. Please find bellow a point-by-point response to the reviewers and their comments.

Reviewer 3

Comments to the editor and authors

The authors isolated and characterized Klebsiella aerogenes from hospitalized patients in Brazil. Considering the severe health impacts of the virulent and antibiotic resistant K. aerogenes, the current study is highly relevant to the public. The authors did a great job in explaining the details on the significance of the research output. Minor corrections should be addressed prior to publication. Those are listed below

  1. Correct spelling - Line 23 (high), Figure 2A (excluded)

Response: We appreciate the reviewers' constructive comments to improve our manuscript. We have revised the sentence (line 23).

  1. Please clarify - what is meant as literature searching in Fig 2A caption; did the authors mean hospital record search or from articles that were previously published?

Response: We thank the reviewer for the comment and apologize for the confusion. The information was revised and corrected in the legend of figure 2.

  1. Please eliminate repeating sentences in lines 182-183.

Response: The sentence was revised following the reviewer’s suggestion (lines 197-198).

  1. Please give the full form for SNP and ESBL wherever it was first given in the text.

Response: We thank the reviewer for the comment and have corrected the terms (lines 120 and 211).

  1. Where any other bacterial agents isolated from the patients from the clinical samples (along with K. aerogenes)? How was it confirmed that the cause of death was solely due to K. aerogenes?

Response: We appreciate the reviewers’ constructive comments and apologize for the confusion. When evaluating the death certificates, we identified the mortality associated with a bacterial infection, we cannot say that the cause of death was due solely to the K. aerogenes isolate, we report the mortality of these patients. In response to the reviewer’s comment, we have included the following statements in the Materials and Methods to explicitly state this limitation in our study (lines 175-178).

“Data regarding the clinical outcome was reviewed. Death due to any cause or death attributable to infection was assessed. Septic shock was defined as sepsis associated with organ dysfunction, accompanied by persistent hypotension following volume replacement.”

  1. Please consider adjusting space between the subtitles and a new paragraph (eg. Line 108 and 109)

Response: Following the reviewer's suggestion, we adjusted the space between the subheadings and a new paragraph.

  1. Please explain Figure 1A – what does the number in the first column represent (before ST)?

Response: We apologize for the confusing wording and we have made these suggested changes to Figure 1.

  1. Consider moving the hypothesis from the results and discussion to the introduction section.

Response: In response to the reviewer’s comment, the sentence was revised and modified (lines 57-58).

Your Sincerely,

Simone Simionatto, PhD